# Weighted Gene Co-Expression Network Analysis Reveals Hub Genes for Fuzz Development in *Gossypium hirsutum*

**DOI:** 10.3390/genes14010208

**Published:** 2023-01-13

**Authors:** Yang Jiao, Yilei Long, Kaixiang Xu, Fuxiang Zhao, Jieyin Zhao, Shengmei Li, Shiwei Geng, Wenju Gao, Peng Sun, Xiaojuan Deng, Quanjia Chen, Chunpin Li, Yanying Qu

**Affiliations:** 1College of Agriculture, Xinjiang Agricultural University, Urumqi 830052, China; 2Institute of Cash Crops, Xinjiang Academy of Agricultural Sciences, Urumqi 830091, China; 3Xinjiang Academy of Agricultural Reclamation, Shihezi 832000, China; 4Xinjiang Kuitun Agricultural and Rural Bureau, KuiTun 833200, China

**Keywords:** WGCNA, fuzz, hub genes, RNA-Seq, *Gossypium hirsutum*

## Abstract

Fuzzless *Gossypium hirsutum* mutants are ideal materials for investigating cotton fiber initiation and development. In this study, we used the fuzzless *G. hirsutum* mutant Xinluzao 50 FLM as the research material and combined it with other fuzzless materials for verification by RNA sequencing to explore the gene expression patterns and differences between genes in upland cotton during the fuzz period. A gene ontology (GO) enrichment analysis showed that differentially expressed genes (DEGs) were mainly enriched in the metabolic process, microtubule binding, and other pathways. A weighted gene co-expression network analysis (WGCNA) showed that two modules of Xinluzao 50 and Xinluzao 50 FLM and four modules of CSS386 and Sicala V-2 were highly correlated with fuzz. We selected the hub gene with the highest KME value among the six modules and constructed an interaction network. In addition, we selected some genes with high KME values from the six modules that were highly associated with fuzz in the four materials and found 19 common differential genes produced by the four materials. These 19 genes are likely involved in the formation of fuzz in upland cotton. Several hub genes belong to the arabinogalactan protein and GDSL lipase, which play important roles in fiber development. According to the differences in expression level, 4 genes were selected from the 19 genes and tested for their expression level in some fuzzless materials. The modules, hub genes, and common genes identified in this study can provide new insights into the formation of fiber and fuzz, and provide a reference for molecular design breeding for the genetic improvement of cotton fiber.

## 1. Introduction

Cotton fiber can be divided into long fiber and fuzz according to the shape of the two types of fiber. From the day of anthesis to two days after anthesis, the epidermal cells develop into long hairs, generally extending to 2.5–3.5 cm; 3–7 days after flowering, epidermal cells develop into fuzzless fibers, usually about 0.5 cm long [1]. During ginning, the long fibers are easily rolled off the seeds, while the fuzz remains [2]. The coexistence of fuzz and fiber in ovules has long been a research hotspot.

Trichomes are highly specialized cells that originate from the epidermal surface of plant leaves, stems, petals, and seeds [3,4]. They contribute to various important functions of plant growth, such as protecting plants against pathogens, insects, and even herbivores, regulating water, and improving tolerance to extreme high temperatures and ultraviolet radiation [5,6]. Many studies have established a foundation for the molecular genetic mechanisms and regulatory networks related to trichome formation. Meanwhile, a large number of functional genes have been identified that regulate epidermal cell development patterns in Arabidopsis, rice, maize, tomato, and artemisia annua; for example, the MYB-bHLH-WD40(MBW) complex [7,8], OsHL6 [9], OsWOX3 [10], ZmOCL4 [11], ZmSPL10/14/26 [12], S1MYC1 [13], and S1Wo [14].

Cotton fibers are single-celled and unbranched trichomes that grow from seed epidermal cells and continuously undergo highly polarized elongation similar to that of leaf trichomes [15,16]. Therefore, the elongation pattern of Arabidopsis trichomes may provide a useful framework for understanding cotton fiber initiation and elongation. It has been reported that members of the lineal homologues of the Arabidopsis MBW complex also play a key role in cotton fiber formation [17]. In addition, recent studies have identified multiple genes that regulate fiber initiation and development; GhPIN3a [18,19] GhARF2/18 [20], and GhHD1 [21,22] are key components of cotton fiber initiation regulation that respond to various plant hormones. In addition, GaDEL65 is mainly related to fuzz initiation and may interact with GhMYB2/3 and GhTTG3 to regulate cotton fiber elongation [23].

Before the sequencing of the cotton genome, researchers mainly screened genes related to cotton fiber development through gene chip technology to understand the mechanism of fiber development at the molecular level. Xu [24] used cDNA chip technology to analyze E6, lipid transfer protein (LTP), proline-rich protein (PRP), expansin, tubulin, annexin, and other gene families related to cotton fiber development, which were found to be expressed differently in the ovules of upland cotton Xuzhou 142 and its long-hairless and short-hairless mutants after flowering. With the completion of cotton genome sequencing, many transgenic technologies have gradually been improved, making it possible to analyze genes and metabolic pathways related to fiber development through the transcriptome [25,26,27], and to identify some candidate genes related to fiber development and the regulation of cell wall deposition and strength [28]. Wang [29] analyzed the 0 DPA, +3 DPA, +5 DPA, and +8 DPA ovules of two Asian cotton materials, and found a total of 3780 differentially expressed genes at the four fiber development stages of the fuzzless mutant (FZ) and wild type (fz), among which 0 DPA had the fewest. With the prolongation of ovular development time, the number of differentially expressed genes gradually increases. KEGG analysis has revealed that these genes are mainly involved in wax and keratin biosynthesis, phenylpropane metabolism, and plant signal transduction. Co-expression trend analysis has shown that among the differentially upregulated genes of the mutant at +3 DPA, the expression of the genes involved in ion binding, MAPK cascade, REDOX activity, and transcriptional regulation is positively affected (increased expression level), resulting in the failure of the normal initiation of mutant fuzzless fibers. These results indicate the dynamic changes of fuzz initiation in diploid Asian cotton, which can provide a reference for the further study of fiber development in *G. hirsutum*.

Weighted gene co-expression network analysis (WGCNA) is based on microarray or RNA-Seq expression data; it is used to construct a scale-free topological overlap matrix through power exponential weighting to describe and analyze the relationship between genes. In this way, genes with similar expression patterns are divided into gene expression modules [30]. WGCNA is mainly used to study the biological correlation between co-expressed gene modules and target traits, and to explore the hub genes in a gene co-expression network. As a typical systems biology method, WGCNA has been widely used in botanical research. Feng et al. [31] showed that the MEmagenta module was highly correlated with fuzzy/fuzzless traits through a weighted gene co-expression network analysis, which included 50 hub genes that were differentially expressed between the two materials. The DEGs were mainly enriched in the energy metabolism and accumulation and auxin signaling pathways at the initiation and extension stages [32]. At the same time, 29 hub genes, including 14-3-3ω, TBL35, GhACS, PME3, GAMMA-TIP, and PUM-7, were identified, among which DEGs and hub genes revealed genetic and molecular mechanisms and differences in cotton fiber development.

Although the transcriptome analysis described above provides some research basis for understanding the initiation process of cotton fiber, the study of key genes in the early stage of fiber development is still in the initial phase, and there are few studies on the dynamic fuzz initiation process.

Fuzzless cotton mutants are important germplasm resources and play an important role in the study of seedless cotton development. With the release of high-quality sequencing data of various cotton species, bioinformatics-based analysis has also facilitated the mapping of functional genes and the exploration of network regulation. GhMML3 and GhMML4 are believed to be responsible for the absence of fuzz or fiber phenotypes in tetraploid fiber mutant lines [33,34,35,36,37,38]. The material used in this study was the tetraploid upland cotton fuzzless mutant Xinluzao 50 FLM, which has fibers but no fuzz (its photoseed traits are controlled by a single dominant gene, with genotype FZ), and the wild-type Xinluzao 50, which has both fibers and fuzz (genotype fz). The mutant and wild-type materials were almost indistinguishable in morphology and development, except that the photoseed mutant had no fiber on the seed coat. Transcriptome sequencing technology was used to analyze the differential gene expression of the mutant and wild type at 0 DPA, +1 DPA, and +3 DPA stages, and a weighted co-expression gene network analysis was performed. The results of the two WGCNAs were combined with the WGCNA of the transcriptome data of other fuzzless materials for validation. The aim was to explore the dynamic process and key genes of fuzz initiation in upland cotton more effectively, and to provide a solid foundation for understanding its molecular mechanism.

## 2. Materials and Methods

### 2.1. Materials

The wild type of upland cotton is Xinluzao 50 (fz), and the fuzzless mutant is Xinluzao 50 FLM (FZ) (Figure 1). These materials were obtained from the Key Laboratory of Biotechnology, Xinjiang Agricultural University, and provided by the Economics Institute of Xinjiang Academy of Agricultural Sciences. In 2019, they were planted at the experimental base of Xinjiang Agricultural University, Shawan County, Xinjiang Uygur Autonomous Region.

Transcriptome data of *G. hirsutum* CSS386 and Sicala V-2 at 2 DPA, 4 DPA, and 6 DPA can be downloaded from https://doi.org/10.25919/5ee196cce855a (accessed on 28 January 2022).

### 2.2. RNA Extraction, Library Construction, and Sequencing

The sampling started on 30 June 2019. Ovule and fiber samples were taken on the day of flowering, or 0 days post anthesis (DPA), and at 1 DPA and 3 DPA, at 11:00 a.m. every day. Each sample was repeated 3 times and immediately put into a foam box with ice bags. Fiber samples were placed in liquid nitrogen storage for subsequent RNA-Seq and qRT-PCR validation.

Total RNA was extracted from frozen cotton fiber tissues using Trizol reagent (Tiangen dp411, Beijing, China). RNA degradation and contamination were examined by 1% agarose gel electrophoresis, and NanoDrop 2000 (Thermo Fisher, Shanghai, China) was used for concentration detection. Samples with od260/280 of 1.8–2.2 and od260/230 of 1.8–2.2 were qualified. RNA with qualified detection quality was sent to Beijing Baimaike Biological Company for cDNA library construction. The cDNA library was sequenced using an Illumina high-throughput sequencing platform to obtain original data. Three biological replicates were performed for each sample of this experiment.

### 2.3. RNA-Seq Data Analysis

First, the fastq-dump software was used to convert downloaded SRA files into FASTQ format documents. Then, FastQC was used to evaluate the quality of sequencing results, the Trimmomatic software(V0.32) [39] (Bolger AM, London, UK) was used for quality control, and the obtained clean data were used for subsequent analysis. Taking the upland cotton genome as the reference gene set [40], the raw counts of each gene in each sample were obtained by using HISAT2 [41] for sequence comparison and the feature ReCounts count [42]. Data were imported into R, and edgeR [43] was used for differential expression analysis. In this study, we selected |log2FC| > 1 (log2FC on behalf of the processing and control expression amount compared to the value of the value) and *p*adj < 0.001 (*p*adj representative of adjusted *p*-values) for the differentially expressed genes.

### 2.4. Construction of Weighted Gene Co-Expression Network

The weighted gene co-expression network was used to analyze the differently expressed genes of Xinluzao 50 and Xinluzao 50 FLM at three different periods. The WGCNA software package [30] in R was used to construct the network. Using the normalized gene expression matrix as the input, a total of 18 transcriptome samples (3 time points, 2 varieties, each with 3 replicates) were identified. After threshold screening, β = 8 was selected to power the original scaled relation matrix to obtain the unscaled adjacency matrix. For a better assessment of the gene expression pattern of correlation among the factors, the topological adjacency matrix can be converted into an overlap matrix (topological overlap matrix, TOM), and by using a topological dissimilarity matrix (dissTOM = 1 − TOM) with a dynamic shear genetic clustering algorithm and module partition. The minimum number of genes in the module was 30 (minModuleSize = 30).

At the same time, the weighted gene co-expression network was analyzed for differentially expressed genes generated by the transcriptomes of the fuzzless upland cotton material CSS386 and the lint Sicala V-2 material at different time points (2, 4, and 6 DPA). Using the normalized gene expression matrix as the input, a total of 18 transcriptome samples (3 time points, 2 varieties, each with 3 replicates) were identified. After threshold screening, β = 10 was selected to power the original scaled relation matrix to obtain the unscaled adjacency matrix.

### 2.5. Screening of Specific Modules Associated with Fuzz

A module is a collection of genes with highly similar expression patterns. To further study the modules associated with tissue height, the module eigengene (ME) and correlation coefficients between different tissues were calculated. To screen specific modules related to fuzz development, the correlation coefficient R and corresponding *p*-values between the feature vectors of each module and different traits (here, the number of days of fiber development) were calculated. R > 0 represents a positive correlation and R < 0 represents a negative correlation. In order to screen modules that were highly correlated with tissue development, in this study we defined the correlation coefficient threshold as 0.60 and the significance level as 0.005, that is, any module with a correlation coefficient higher than 0.60 and a significance level of 0.005 was considered as a specific module for fuzz development [44].

### 2.6. Enrichment Analysis

First, genes in modules significantly associated with short hair development were extracted, and then OmicShareTools (https://www.omicshare.com/tools/, accessed on 28 January 2022, Guangzhou, China) was used for GO and KEGG enrichment analysis. The thresholds were *p* < 0.01 and *Q* < 0.05.

### 2.7. qRT-PCR Validated Transcriptome Sequencing

We analyzed the RNA-Seq data to verify the accuracy of the transcriptome data. Eight differentially expressed genes were randomly selected and verified by qRT-PCR. The NCBI primer website (https://www.ncbi.nlm.nih.gov/tools/, accessed on 28 January 2022) was used to design gene-specific primers (Appendix A). A FastKing RT kit (Tiangen KR116, Beijing, China) was used for reverse transcription. A PerfectStart Green qPCR SuperMix kit (TransGen BiotechAQ601, Beijing, China) was used for quantitative amplification with a volume of 20 µL. The reaction mix contained 10 µL 2 × PerfectStart Green SuperMix, 2 µL template cDNA, 0.4 µL F primer (10 µM), 0.4 µL R primer (10 µM), 0.4 µL passive reference dye (50×), and 6.8 µL nuclease-free water. The qRT-PCR program was run in two steps using the ABI 7500 Fast Real-Time PCR System to complete the amplification as follows: pre-denaturation at 94 °C for 35 s, followed by 35 cycles of denaturation at 94 °C for 5 s, then fluorescence signal at 60 °C for 30 s. The experiments were performed in triplicate, the internal reference gene was UBQ7, and the relative gene expression levels were calculated using the 2^−ΔΔCt^ method.

## 3. Results and Analysis

### 3.1. Data Processing and Analysis

In order to further understand the dynamic process of fuzz development and identify related genes, mutant Xinluzao 50 FLM and wild-type Xinluzao 50 materials were selected to extract RNA from ovules and fibers at 3 developmental stages (0 DPA, +1 DPA, +3 DPA) with 3 biological duplicates for each material, and 18 RNA libraries were constructed (Appendix A).

According to further PCA, in the Xinluzao 50 and Xinluzao 50 FLM materials (Figure 2), polymerization was generally the same in the different development periods. At the early stage of fiber development, the polymerization degree of 0 DPA and 1 DPA was high and the trend was roughly the same, which may be because the fiber and ovule were not completely separated at these stages. When the ovule developed to 3 DPA, the aggregation trend of wild-type and mutant material changed, probably because the differentiation of short velvet started at this stage. Similarly, in CSS386 and Sicala V-2 (Figure 3), samples of different materials from the same period were gathered together, and there were no significant differences between the three repeatable samples in each group. The PCA results further confirmed the relationship between the two materials at different fiber development stages.

### 3.2. Differential Gene Expression Analysis of FZ and fz

After calculating the FPKM value, we selected a *p*-value < 0.05 and |log2(ratio)| ≥ 1 or more as a standard for screening differentially expressed genes. A total of 12,009 differential genes were identified by comparing the FPKM values of FZ materials without fuzz and wild-type FZ materials at 0, +1, and +3 DPA. There were 3812, 8547, and 1655 differentially expressed genes in the wild-type and mutant materials at 0, 1, and 3 DPA, respectively. Among these, the three periods produced 257 common genes (Figure 4A). Similarly, 6175, 4925, and 12,889 differentially expressed genes were generated by comparing the DPA of CSS386 without fuzz material and Sicala V-2 at 2, 4, and 6 DPA (Figure 4B). Among these, the three periods produced 739 common genes (Figure 4).

### 3.3. Functional Analysis of Differential Genes

To assess the major biological and regulatory functions of the identified differential genes, we performed GO enrichment analysis on the differential genes involved in each developmental period. In the three fiber development periods of Xinluzao 50 and Xinluzao 50 FLM, at 0 DPA, in biological processes, they were mainly enriched in translation (GO:0006412), the peptide biosynthetic process (GO:0043043), and the peptide metabolic process (GO:0006518); in cellular components, they were mainly enriched in the nucleosome (GO:0000786), DNA packaging complex (GO:0044815), and protein–DNA complex assembly (GO:0065004); and in molecular functions, they were mainly enriched in structural molecule activity (GO:0005198), the structural constituent of the ribosome (GO:0003735), and protein heterodimerization activity (GO:0046982) (Appendix A). At 1 DPA, in biological processes, they were mainly enriched in the cellular macromolecule metabolic process (GO:0044260), gene expression (GO:0010467), and the cellular nitrogen compound metabolic process (GO:0034641); in cellular components, they were mainly enriched in the cell part (GO:0044464), macromolecular complex (GO:0032991), and ribonucleoprotein complex (GO:0030529); and in molecular functions, they were mainly enriched in the structural constituent of the ribosome (GO:0003735), structural molecule activity (GO:0005198), and RNA binding (GO:0003723) (Appendix A). At 3 DPA, in biological processes, they were mainly enriched in translation (GO:0006412), the peptide biosynthetic process (GO:0043043), and the peptide metabolic process (GO:0006518); in cellular components, they were mainly enriched in the mitochondrial respiratory chain (GO:0005746), respiratory chain complex (GO:0098803), and respiratory chain (GO:0070469); and in molecular functions, they were mainly enriched in the structural constituent of the ribosome (GO:0003735), structural molecule activity (GO:0005198), and snoRNA binding (GO:0030515) (Figure 5).

In the three fiber development stages of fuzzless material CSS386 and fuzz material Sicala V-2, at 2 DPA, in biological processes, they were mainly enriched in cellular potassium ion homeostasis (GO:0030007), potassium ion homeostasis (GO:0055075), and the glutamine family amino acid catabolic process (GO:0009065); in cellular components, they were mainly enriched in the endocytic vesicle (GO:0030139), vacuole (GO:0005773), and cell surface (GO:0009986); and in molecular function, they were mainly enriched in C4-dicarboxylate transmembrane transporter activity (GO:0015556), dicarboxylic acid transmembrane transporter activity (GO:0005310), and arginine decarboxylase activity (GO:0008792) (Appendix A). At 4 DPA, in biological processes, they were mainly enriched in microtubule-based movement (GO:0007018), movement of the cell or subcellular component (GO:0006928), and external encapsulating structure organization (GO:0045229); in cellular components, they were mainly enriched in the microtubule (GO:0005874), supramolecular fiber (GO:0099512), and polymeric cytoskeletal fiber (GO:0099513); and in molecular function they were mainly enriched in microtubule binding (GO:0008017), microtubule motor activity (GO:0003777), and tubulin binding (GO:0015631) (Appendix A). At 6 DPA, in biological processes, they were mainly enriched in the regulation of the macromolecule metabolic process (GO:0060255), primary metabolic process (GO:0044238), and cellular metabolic process (GO:0044237); in cellular components, they were mainly enriched in the vacuolar membrane (GO:0005774), vacuolar part (GO:0044437), and nucleus (GO:0005634); and in molecular function, they were mainly enriched in cellulose synthase activity (GO:0016759), cellulose synthase (UDP-forming) activity (GO:0016760), and hydrolase activity (GO:0016787) (Appendix A).

### 3.4. Consistency Evaluation of qRT-PCR and RNA-Seq

In order to confirm the reliability of our RNA sequences, eight genes were randomly selected for qRT-PCR experiments (Appendix A). It was found that they were basically consistent with the trend of the transcriptome expression profile, which proved that the transcriptome data were reliable for the next analysis (Figure 6).

### 3.5. Construction of Weighted Gene Co-Expression Network

A total of 31,216 gene expression profiles were obtained through data analysis. Comparing the differential genes of different materials, Xinluzao 50 FLM and Xinluzao 50 were found to produce 12,009 differential genes (Figure 7A), and CSS386 and Sicala V-2 produced 19,207 differential genes in the three periods. We filtered the genes with low expression (Figure 7B) and selected FPKM > 1 to construct the co-expression network.

Using the dynamic pruning tree method to combine the weight values and express similar modules, 12 modules were obtained in Xinluzao 50 and Xinluzao 50 FLM materials. The turquoise module contained the largest number of genes, 2982 genes, and the tan and green–yellow modules contained the smallest number of genes, only 36 genes, with an average of 905 genes per module (Figure 8A).

A total of 10 co-expression modules were obtained from the fuzzless CSS386 and Sicala-2 materials, and different colors were used to represent different modules, with each module containing a different number of genes. Among them, the turquoise module contained the largest number of genes, 10,061 genes, and the purple module contained the smallest number of genes, only 55 genes, with an average of 1361 genes per module (Figure 8B).

### 3.6. Identification of Specific Modules Related to Fuzz

In the Xinluzao 50 and Xinluzao 50 FLM materials, according to the criteria of |r| > 0.60 and *p* < 0.005, there was a significant positive correlation between the four modules and the development of fuzz (Figure 9). For example, MEyellow (r = 0.93, *p* < 0.000000018) was highly correlated with 3 DPA, which is the key stage of fuzz development. MEred (r = 0.68, *p* < 0.0021) was also correlated with 3 DPA. In addition, MEyellow (r = 0.66, *p* < 0.0031) was highly correlated with fuzz.

CSS386 and Sicala V-2, according to |r| > 0.70 and *p* < 0.001, showed four modules with a significant positive correlation with the development of fuzz (Figure 10). For example, MEred (r = 0.97, *p* < 7 × 10^−11^) was highly correlated with 4 DPA, which is the key stage of fuzz development. MEgreen (r = 0.77, *p* < 0.00019) was highly correlated with fuzz development.

MEyellow and MEred were selected from Xinluzao 50 and Xinluzao 50 FLM materials for analysis. In CSS386 and Sicala V-2 materials, MEred, MEgreen, MEblue, and MEpink were selected for analysis.

### 3.7. Functional Enrichment Analysis of Development-Specific Modules with Fuzz

In order to screen the genes related to the fuzz development of upland cotton, in this study, we enriched and analyzed the module genes significantly associated with fuzz development. MEred and MEyellow modules were selected for the KEGG enrichment analysis of Xinluzao 50 and Xinluzao 50 FLM (Figure 11), and MEgreen, MEred, MEblue, and MEpink modules were selected for the analysis of fuzzless CSS386 and Sicala V-2 materials (Figure 12).

In Xinluzao 50 and Xinluzao 50 FLM, MEred and MEyellow were selected for analysis. The MEyellow module was mainly enriched in ribosomes (ko03010), oxidative phosphorylation (ko00190), proteasome (ko03050), and other pathways; the MEred module was mainly enriched in the MAPK signaling pathway (ko04016), synthesis and degradation of ketone bodies (ko00072), and other pathways. In the CSS386 and Sicala V-2, MEred, MEgreen, MEblue, and MEpink were selected for analysis. MEred was mainly enriched in fatty acid elongation (ko00062), the biosynthesis of secondary metabolites (ko01110), Cutin, suberin, and wax biosynthesis (ko00073). MEgreen was enriched in tyrosine metabolism (ko00350), starch and sucrose metabolism (ko00500), glycolysis/gluconeogenesis (ko00010), and other pathways. MEblue was mainly enriched in flavonoid biosynthesis (ko00941), phenylalanine, tyrosine and tryptophan biosynthesis (ko00400), the biosynthesis of secondary metabolites (ko01110), fatty acid degradation (ko00071), and other pathways. MEpink was mainly enriched in fatty acid metabolism (ko01212), fatty acid elongation (ko00062), the biosynthesis of unsaturated fatty acids (ko01040), and other pathways.

### 3.8. Analysis of Hub Genes of Modules Related to Fuzz Development in Four Materials

According to the fuzz development modules generated by the four materials, we select hub genes with a gene significance (GS) greater than 0.80 and a module connectivity (KME) value greater than 0.9.

In Xinluzao 50 and Xinluzao 50 FLM materials, 669 and 677 hub genes were generated by MEyellow and MEred. In the CSS386 and Sicala V-2 materials, 504 hub genes were produced by MEred, MEgreen, MEblue, and MEpink.

In order to explore the genes related to fuzz in the fuzzless mutant Xinluzao 50 FLM, we focused on the MEyellow and MEred modules constructed from the Xinluzao 50 and Xinluzao 50 FLM materials. MEred, MEgreen, MEblue, and MEpink constructed by CSS386 and Sicala V-2 were used for verification. Among the hub genes selected by the 6 modules generated by these 4 materials, 19 were common genes, and these 19 genes may be involved in the development of fuzz in Xinluzao 50 fuzzless materials; they are annotated in Appendix A. Figure 13 shows heat maps of the expression of the 19 genes in the four materials.

As can be seen from the figure, the expression levels of the genes *GH_A01G1670*, *GH_A05G1584*, *GH_A08G1222*, and *GH_D01G2481* were higher than those of other genes in the +3 DPA stage of Xinluzao 50 and its mutant Xinluzao 50 FLM.

### 3.9. qRT-PCR of Hub Genes

Gene *GH_A01G1670* belongs to the LTPG family and plays an important role in fiber development [45]. *GH_A05G1584* is a multifunctional hydrolase, which is known to be a functional enzyme involved in ovular and fiber development and plays an important role in seed development [46]. *GH_A08G1222* belongs to the BAHD enzyme [47], and *GH_D01G2481* belongs to metallothionein.

The above four genes were selected for fluorescence quantitative PCR detection in some fuzzless materials (Appendix A). As shown in Figure 14, the gene *GH_A01G1670* was compared with TM-1 with fiber and fuzz material. At 3 DPA, the expression levels of n2 and N1 in the fuzzless material were higher than those in TM-1, showing significant differences. Compared with the fuzzless mutants GZNn FLM and Xinluzao 50 FLM of upland cotton, the expression levels of GZNn and Xinluzao 50 were lower, and there was a significant difference. The results showed that *GH_A01G1670* negatively regulated the development of fuzz in upland cotton, but in Asian cotton, the expression level of the homologous *Ga02G0907* gene of this gene in DPL971 with lint was higher than that in DPL972 without fuzz, and the difference was significant. This suggests that the homologous gene positively regulates the development of fuzz. The reason for this phenomenon may be that the mechanism of fuzz development is different between upland cotton and Asian cotton (Appendix A).

This phenomenon was also caused by the gene *GH_D01G2481*. Taking TM-1 with fiber and fuzz as the control, the expression levels of n2 and N1 in TM-1 without fuzz were lower at 3 DPA, showing a significant difference. The expression of GZNn and Xinluzao 50 in upland cotton with lint was higher than that of GZNn FLM and Xinluzao 50 in upland cotton without fuzz. The results show that *GH_D01G2481* positively regulated the development of fuzz in upland cotton. However, in Asian cotton, the expression level of the homologous *Ga02G1745* gene in DPL971 with fuzz was significantly lower than that in the mutant DPL972. This suggests that the homologous gene negatively regulates the development of fuzz (Figure 15, Appendix A).

The gene *GH_A05G1584* was compared with TM-1. At 3 DPA, the expression level of N1 fuzzless material was higher than that of TM-1, and there was a significant difference. The expression of GZNn and Xinluzao 50 in upland cotton with fuzz was lower than that of GZNn FLM and Xinluzao 50 in upland cotton without fuzz. The results showed that *GH_A05G1584* negatively regulated the development of fuzz in upland cotton; *Ga05G1683*, the homologue of this gene, also negatively regulated short down development in Asian cotton (Figure 16, Appendix A). The gene *GH_A08G1222* was compared with TM-1. At 3 DPA, the expression levels of n2 and N1 in the fuzzless material were lower than those in TM-1, and there was a significant difference. GZNn and Xinluzao 50 had a higher expression than GZNn FLM and Xinluzao 50. The results showed that *GH_A08G1222* positively regulated the development of fuzz in upland cotton. In Asian cotton, the homolog of this gene, *Ga08G1181*, also positively regulated the development of fuzz (Figure 17, Appendix A).

### 3.10. Module Hub Gene Mining and Interaction Network Analysis

The top five genes with the largest KME (eigengene connectivity) value were selected as the hub genes of each module. The hub gene *GH_D11G2656* in the MEyellow module was generated by Xinluzao 50 FLM and Xinluzao 50, which belongs to the arabinogalactan protein AGP, the hub gene *GH_A13G2626* in the MEred module belongs to glucosidase, and *GH_A04G1078* belongs to the MYB transcription factor.

The Cytoscape software was used to select the top 20 related genes with weight values for visualization, and an interaction network of the hub genes of the two modules was constructed (Figure 18). The *GH_D09G0176* gene interacting with *GH_A04G1078* is also an MYB transcription factor, which may jointly regulate the development of cotton fuzz [48]. In the hub gene *GH_A04G1689* of the MEred module, the expression of GhLTPG1 in Arabidopsis leads to an increase in trichome number [46]. The genes that interact with this hub gene (*GH_A07G2540*, *GH_A10G0345*, and *GH_A05G3781*) are known from experiments. It is a multifunctional hydrolase related to fiber development [49]. In the MEyellow module, the gene *GH_D11G1443*, which interacts with the hub gene *GH_A13G1990*, belongs to the TCP family and plays an important role in the growth and development of cotton fiber [50]. The gene *GH_D05G1346*, which interacts with the hub gene *GH_A06G2184*, belongs to GDSL lipase and plays an important role in wax and epidermal cell layer biosynthesis [49].

## 4. Discussion

Based on the RNA-Seq data of two pairs of upland cotton varieties, we identified 12,009 and 19,207 differentially expressed genes. GO enrichment analysis of the three fiber development stages of Xinluzao 50 and the mutant Xinluzao 50 FLM showed that at 0 DPA, they were mainly enriched in the nucleosome and DNA packaging complex in cell components; at 1 DPA they were mainly enriched in the genes involved in cellular macromolecular metabolism and cellular nitrogen compound metabolism in biological processes; and at 3 DPA they were mainly enriched in the genes of the mitochondrial respiratory chain and respiratory chain complex in cell components. In the three fiber development stages of CSS386 and Sicala V-2, the molecular functions at 2 DPA were mainly enriched in C4-dicarboxylic acid transmembrane transporter and dicarboxylic acid transmembrane transporter activity; at 4 DPA, the molecular functions were mainly enriched in microtubule binding, microtubule motility, and tubulin binding; and at 6 DPA, they were mainly enriched in the regulation of macromolecular metabolic processes and primary metabolic processes in biological processes.

### 4.1. Hub Genes in the Six Modules May Be Involved in the Formation of Cotton Fuzz

The initiation of fiber and fuzz is a rather complex and ambiguous process. To date, multiple sites and factors have been identified to control or regulate the initiation of fiber and fuzz. Based on a previous analysis, Feng et al. [31] analyzed the gene expression and regulatory network of two Asian cotton varieties with significant differences in villus characteristics. The DEGs identified by the transcriptomic analysis of fiber-attached ovules at 1, 3, and 5 DPA were used to establish a weighted gene co-expression network analysis method based on Asian cotton fuzz initiation and formation. In this study, to explore the genes related to the fuzz mutant Xinluzao 50 FLM, we used the MEyellow and MEred modules constructed from the materials of Xinluzao 50 FLM and Xinluzao 50 as the basis and combined them with the CSS386 and Sicala V-2 materials. Among the hub genes selected by the six modules generated from the four materials, the MEred, MEgreen, MEblue, and MEpink modules indicated that 19 genes were shared genes and may be involved in Xinluzao 50 fuzz development. The highlight of this paper is the combination of the results of the two WGCNAs, and the identification of some genes related to the development of fuzz.

In addition, previous studies have reported that *GhMML3_A12* affects the formation of fuzz in upland cotton N1 [34]. This gene was also identified in the MEgreen module, indicating the reliability of the modules and networks of interest identified from the WGCNA.

The top five genes with maximum KME (eigengene connectivity) value were selected as the hub genes of each module and an interaction network of hub genes, such as *GH_D11G2656* in the MEyellow module generated by Xinluzao 50 and Xinluzao 50 FLM, was constructed. It belongs to the arabinogalactan protein (AGP), which is abundant in fiber development and may be involved in fiber initiation and elongation. GhFLA1 also belongs to AGP, and its overexpression in cotton promotes fiber elongation and leads to increased fiber length [50,51]. *GH_D13G2485*, which interacts with this hub gene, and the hub gene of the MEpink module both belong to the LTPG family and play important roles in fiber development [45]. In the hub gene *GH_A04G1689* of the MEred module, the expression of GhLTPG1 in Arabidopsis leads to an increase in trichome number, while GhLTPG1 knockdown in cotton fibers shows significantly shortened length, reduced polar lipid content, and the expression of genes related to inhibited fiber elongation [45]. The genes interacting with this hub gene (*GH_A07G2540*, *GH_A10G0345*, and *GH_A05G3781*) are multifunctional hydrolases. The GDSL lipid gene (GhGLIP) is obtained from the ovule and fiber in cotton (*G. hirsutum* L. cv Xzhou 142). Experiments have shown that this gene is a functional enzyme involved in ovule and fiber development, and plays an important role in seed development [50].

### 4.2. Some Genes Are Involved in Fuzz Development of Two Kinds of Fuzzless Upland Cotton

Wax inducer 1 (WIN1) was first reported to transcriptionally activate epidermal wax biosynthesis in Arabidopsis [52]. Tomato (*Solanum lycopersicum*) SlWIN3/SHN3 genes regulate the formation of the cuticle in fleshy fruits [53]. The cuticle is typically a feature of cells exposed to air at some point in their life cycle. WIN transcription factors regulate flower organ epidermis in Arabidopsis [54]. In cotton fibers, a mixture of fats, waxes, and resins is released onto the cell surface during maturation. The cuticle is less than 0–25 microns thick, and although tightly molded to the main wall, remains intact except in the most serious fiber defects. The development of fuzz may be closely related to the formation of the cuticle.

GDSL lipase is an important subclass or subfamily of lipolyases and is widely present in microorganisms [55]. In plants, GDSL lipase has been found to be a multifunctional enzyme involved in many physiological processes, including plant growth, morphogenesis, and pathogen responses [56,57,58,59,60]. The first plant GDSL lipase gene was isolated from *Brassica napus* [61], and 105, 121, and 114 GDSL esterase/lipase genes were reported in Arabidopsis, Brassica, and rice, respectively [62,63,64]. The Arabidopsis GDSL lipase genes GLIP1 and GLIP2 play important roles in regulating systemic resistance and pathogen defense, respectively [65,66]. Rice GER1 is a GDSL motif-encoding gene that regulates germ elongation through light–JA interaction. WDL1, a member of the GDSL lipase family, plays an important role in rice wax and epidermal cell layer biosynthesis [67]. Pepper GDSL lipase plays an important role in abiotic stress and pathogen defense [58,68]. The DSL1 protein of tomato is required for the extracellular deposition of keratinized polyester in the fruit cuticle [69]. Several BAHD enzymes are important for the correct synthesis or assembly of keratin, suberin, and related waxes [47].

The nsLTP gene was isolated from cotton fibers several decades ago [70]. GH3, Ltp3, Ltp6, and GhLTPG1 were confirmed to be specifically expressed in fibrocytes, and Ltp3 was shown to reach its maximum expression at the late stage of fiber elongation [45,70,71]. An expression analysis showed that GhLtp6, GhLtp7, GhLtp8, and GhLtp11 were highly expressed during fiber initiation [70]. Seed trichomes, fiber initiation, and elongation are regulated by the MYB gene [45,72,73,74,75,76]. Cotton LTP3 is regulated by the MYB protein [77]. Therefore, this supports that NSLTP is involved in fiber development.

In this study, 19 common genes were found in the Xinluzao 50, Xinluzao 50 FLM, CSS386 and Sicala V-2 materials by selecting hub genes with high KME values in the WGCNA constructed from four kinds of upland cotton through RNA-Seq. The GDSL lipase and BAHD enzyme may be involved in the formation of the cotton fiber cuticle, and the SHN3 and LTP genes may regulate the formation of cotton fuzz. Among the 19 common genes, some genes with high expression levels during fuzz development were selected for quantitative fluorescence detection in some studies. It is noteworthy that some genes positively regulate fuzz development in upland cotton, but negatively regulate fuzz formation in Asian cotton. This phenomenon may be caused by the different mechanism of fuzz development between the two types of cotton [29].

## 5. Conclusions

In this study, we compared the RNA-Seq data of Xinluzao 50, Xinluzao 50 FLM, CSS386 and Sicala V-2 and performed a GO enrichment analysis on the differential genes in six development periods. Among the differential genes generated by these four materials, we analyzed the weighted gene co-expression network and found six modules highly related to fuzz. We performed a KEGG enrichment analysis on these six modules. We selected the core genes with the highest KME values among the six modules and constructed an interaction network. In addition, among the six modules highly related to fuzz in the four materials, we selected some genes with higher KME values, and found 19 common differential genes generated among the four types. These 19 genes may be involved in the formation of fuzz in upland cotton. Some genes in some fuzz materials were also selected for qRT-PCR. In the future, we can carry out further in-depth research by means of biotechnology approaches such as overexpression, VIGS, and gene knockout. This study revealed the regulation mechanism of fuzz formation and the development stages of upland cotton, and lays a foundation for the molecular breeding of cotton with improved fiber characteristics.

## Figures and Tables

**Figure 1 genes-14-00208-f001:**
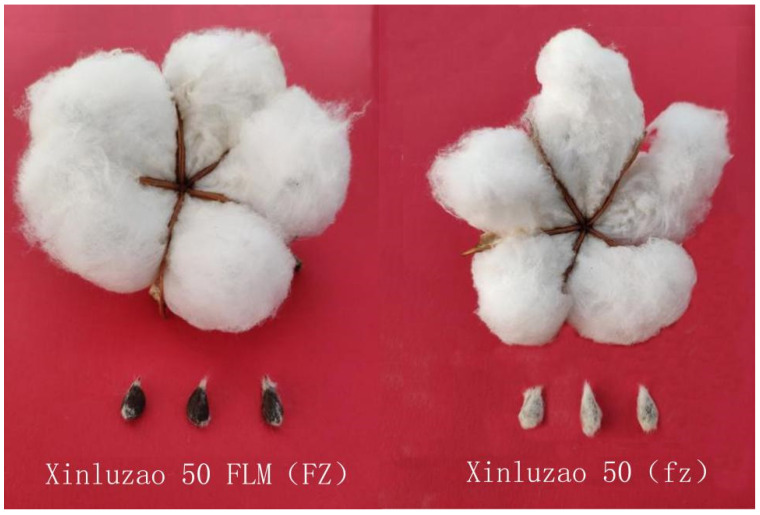
Wild type (fz) and fuzz mutant (FZ).

**Figure 2 genes-14-00208-f002:**
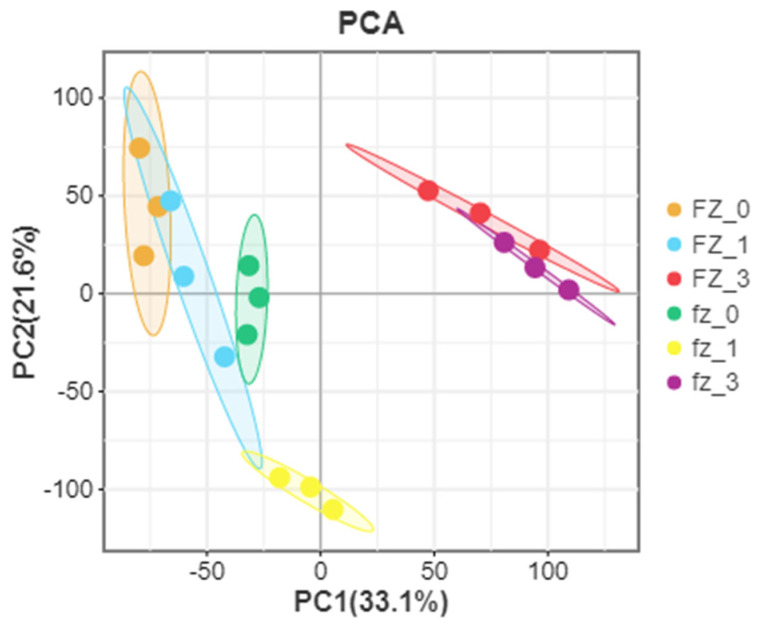
PCA analysis of Xinluzao 50 (fz) and Xinluzao 50 FLM (FZ).

**Figure 3 genes-14-00208-f003:**
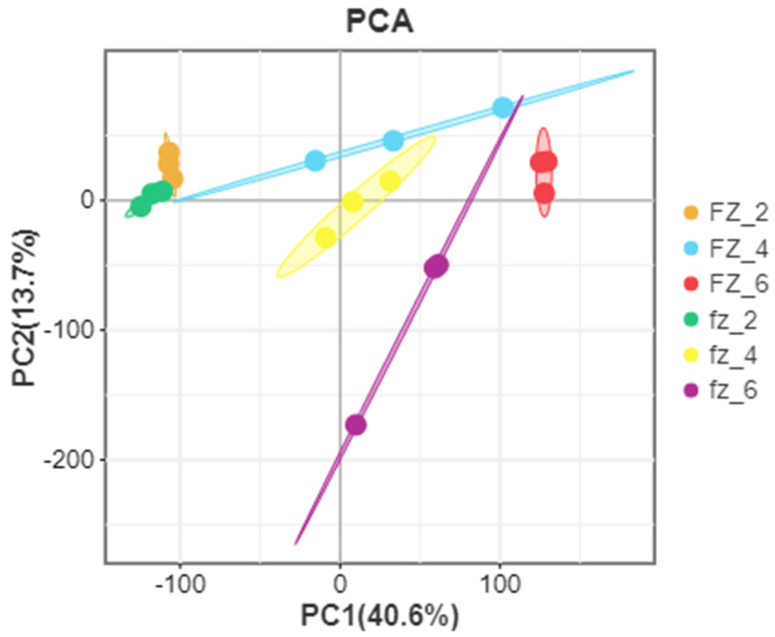
PCA analysis of CSS386 (FZ) and Sicala V-2 (fz).

**Figure 4 genes-14-00208-f004:**
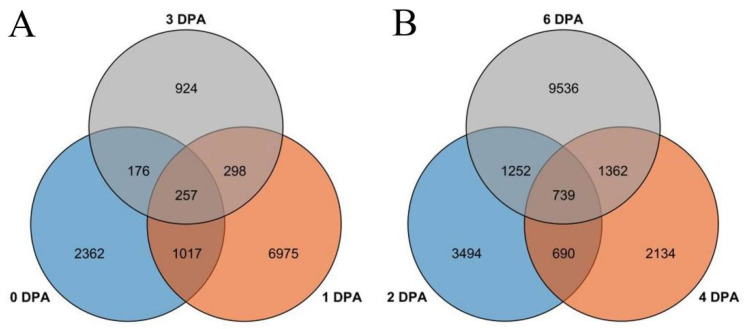
Venn diagram of genes of four materials in different periods: (**A**) Xinluzao 50 and Xinluzao 50FLM; (**B**) CSS386 and Sicala V-2.

**Figure 5 genes-14-00208-f005:**
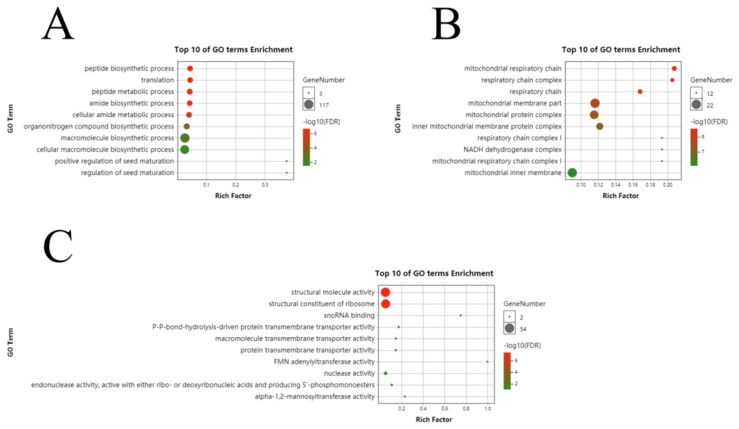
GO analysis of differential genes between Xinluzao 50 and Xinluzao 50 FLM in (**A**) biological processes, (**B**) cellular components, and (**C**) molecular function.

**Figure 6 genes-14-00208-f006:**
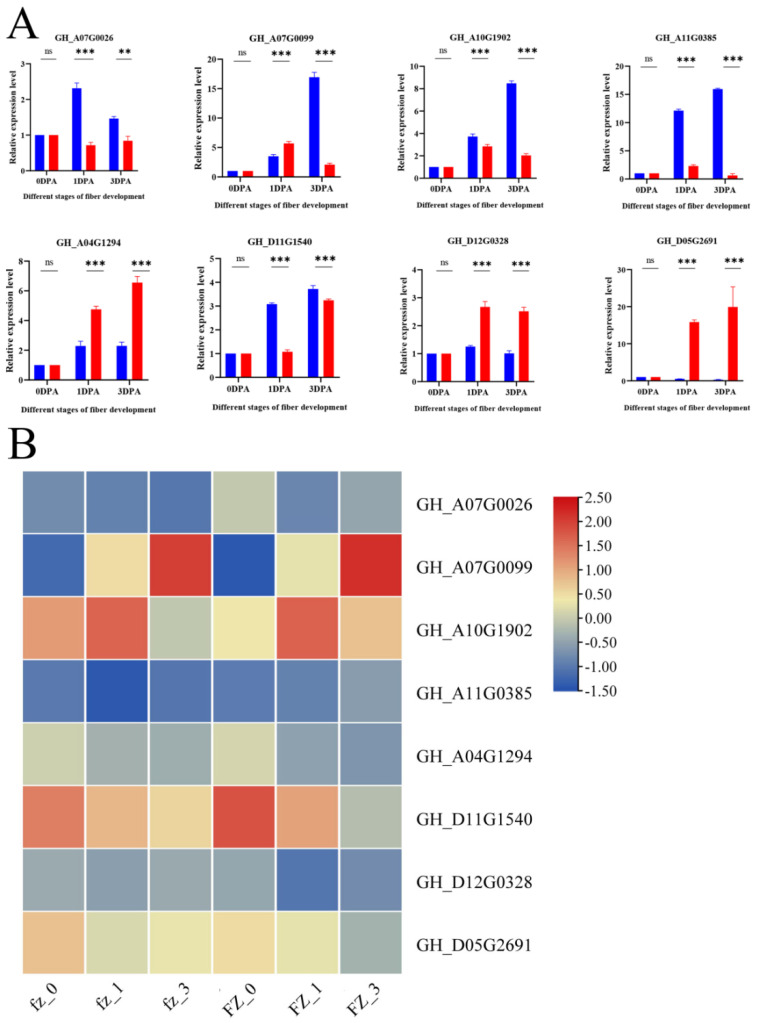
RNA-Seq results were confirmed using qRT-PCR experiments. (**A**) qRT-PCR results of eight genes. Relevant significance level (*p*-value) shown above each histogram. ns indicates no significant effect at *p* < 0.05; ** and *** represent significance at *p* < 0.01 and *p* < 0.001 levels, respectively. Pink bars, L; blue bars, S. (**B**) RNA-Seq heatmap of eight genes.

**Figure 7 genes-14-00208-f007:**
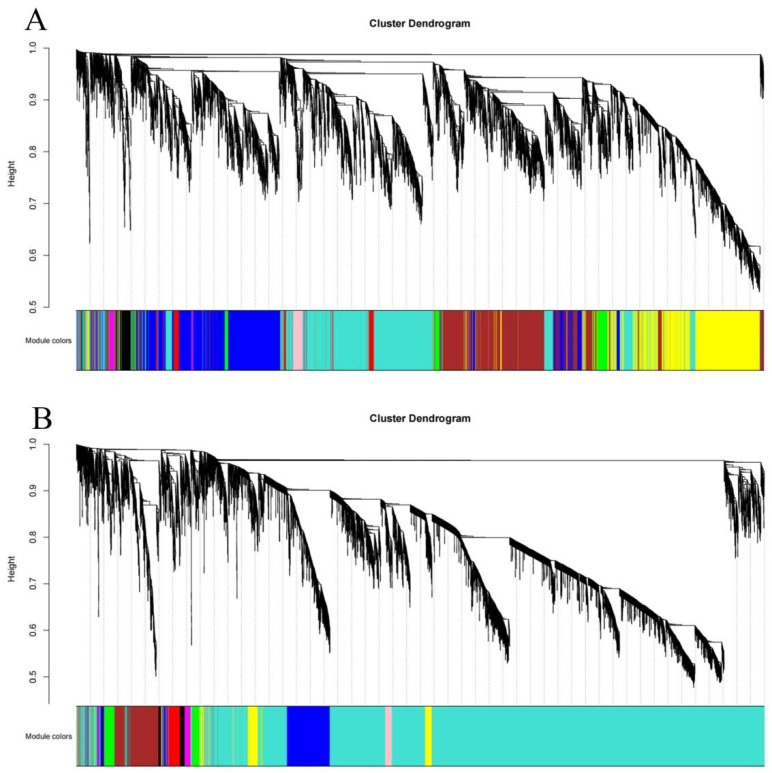
Gene cluster tree and module construction: (**A**) Xinluzao 50 and Xinluzao 50 FLM; (**B**) CSS386 and Sicala V-2.

**Figure 8 genes-14-00208-f008:**
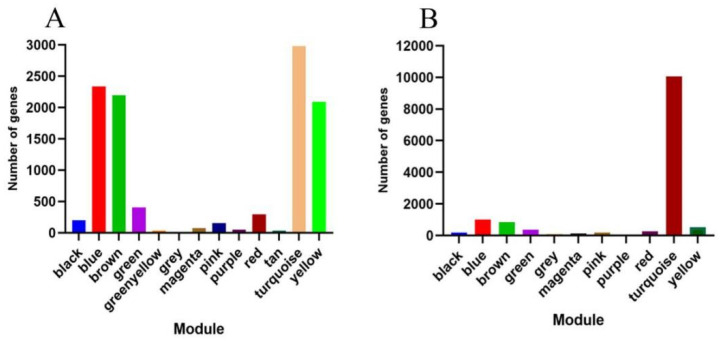
Genes included in different modules: (**A**) Xinluzao 50 and Xinluzao 50 FLM; (**B**) CSS386 and Sicala V-2.

**Figure 9 genes-14-00208-f009:**
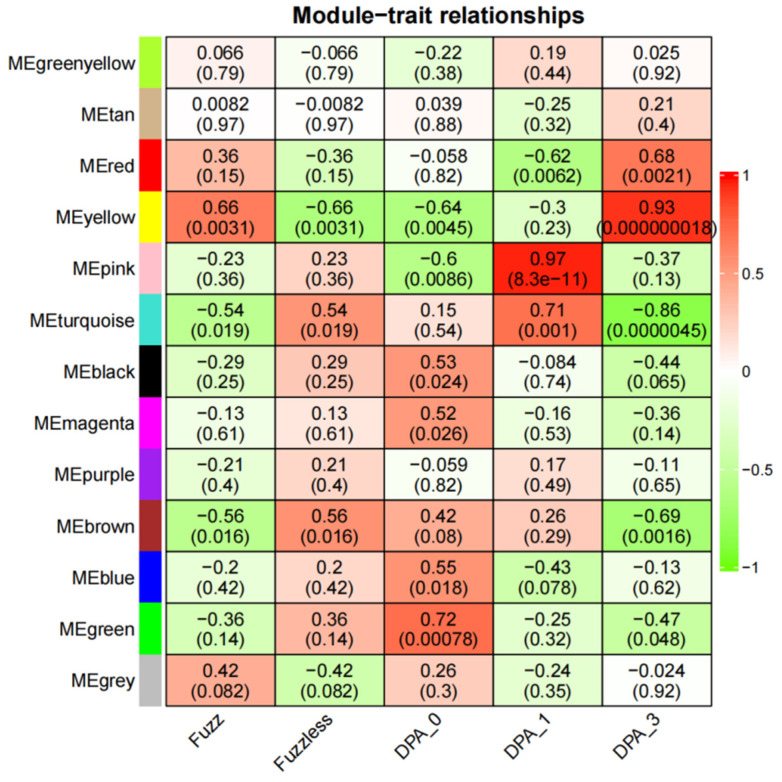
Heatmap of correlation between modules and traits. Numbers in squares represent correlation coefficients and *p*-values between modules.

**Figure 10 genes-14-00208-f010:**
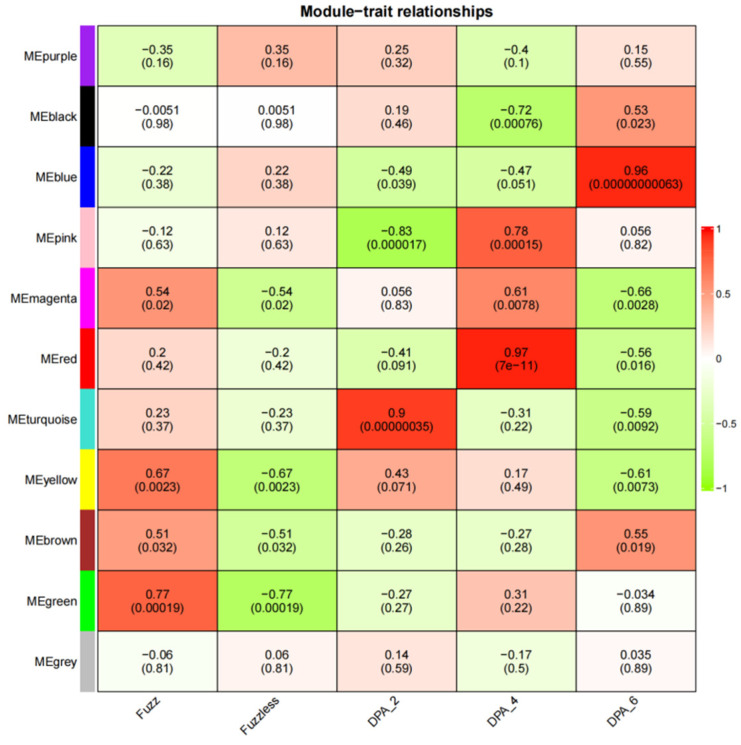
Heatmap of correlation between modules and traits. Numbers in squares represent correlation coefficients and *p*-values between modules.

**Figure 11 genes-14-00208-f011:**
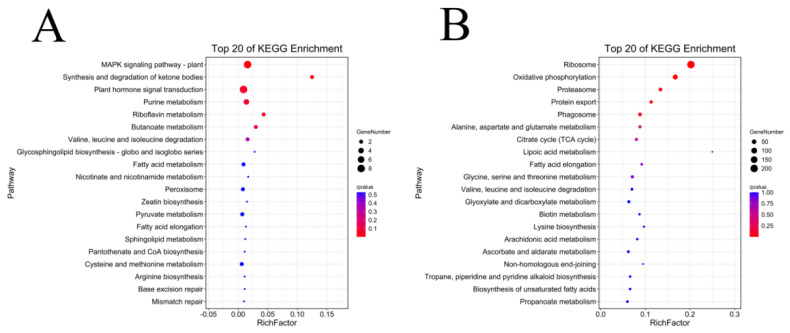
KEGG enrichment analysis of genes in target module (Xinluzao 50 and Xinluzao 50 FLM): (**A**) MEred module; (**B**) MEyellow module.

**Figure 12 genes-14-00208-f012:**
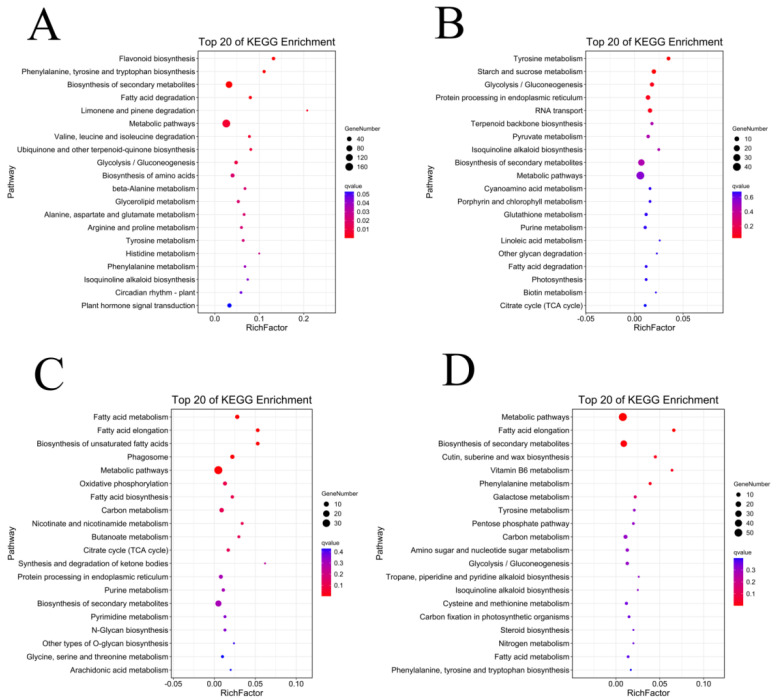
KEGG enrichment analysis of genes in target module (CSS386 and Sicala V-2): (**A**) MEblue module; (**B**) MEgreen module; (**C**) MEpink module; (**D**) MEred module.

**Figure 13 genes-14-00208-f013:**
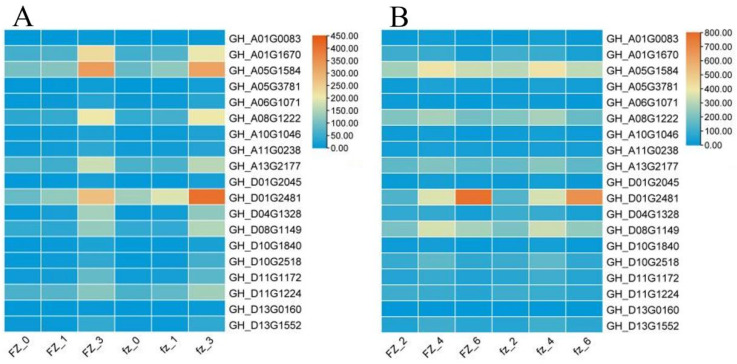
Analysis of hub gene expression: (**A**) Xinluzao 50 and Xinluzao 50 expression heatmaps; (**B**) CSS386 and Sicala V-2 expression heatmaps. FZ, fuzzless material; fz, fuzz material.

**Figure 14 genes-14-00208-f014:**
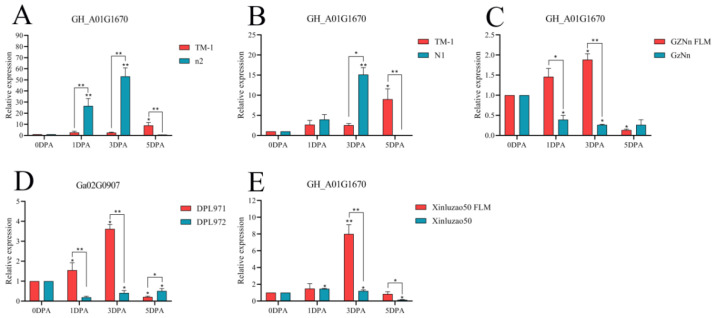
Relative expression levels of the *GH_A01G1670* gene in upland cotton. (**A**–**C**,**E**) Upland cotton; (**D**) homologous gene of Asian cotton. Asterisks at top of bars indicate statistically significant differences between periods (* *p* < 0.05, ** *p < 0.01)*.

**Figure 15 genes-14-00208-f015:**
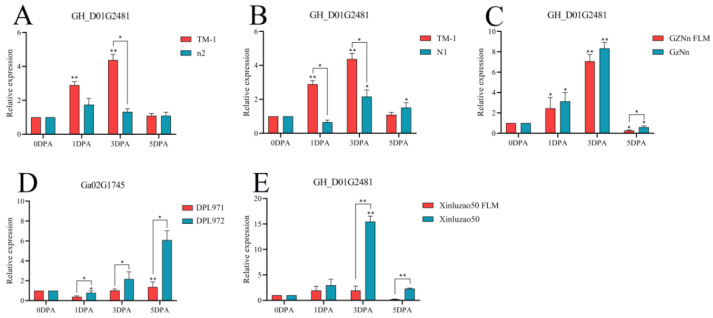
Relative expression levels of the *GH_D01G2481* gene in upland cotton. (**A**–**C**,**E**) Upland cotton; (**D**) homologous gene of Asian cotton. Asterisks at top of bars indicate statistically significant differences between periods (* *p* < 0.05, ** *p* < 0.01).

**Figure 16 genes-14-00208-f016:**
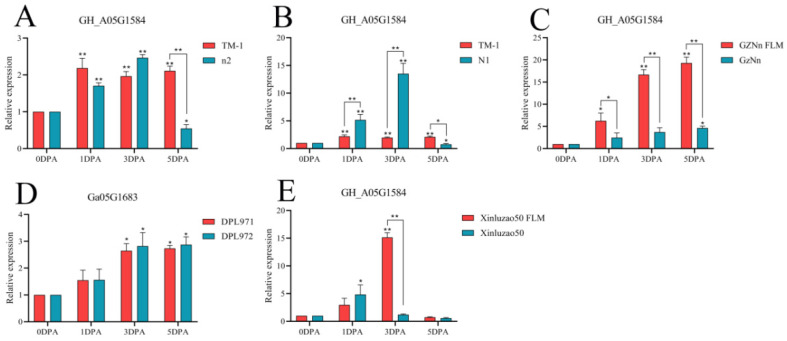
Relative expression levels of the GH_A05G1584 gene in upland cotton. (**A**–**C**,**E**) Upland cotton; (**D**) homologous gene of Asian cotton. Asterisks at top of bars indicate statistically significant differences between periods (* *p* < 0.05, ** *p* < 0.01).

**Figure 17 genes-14-00208-f017:**
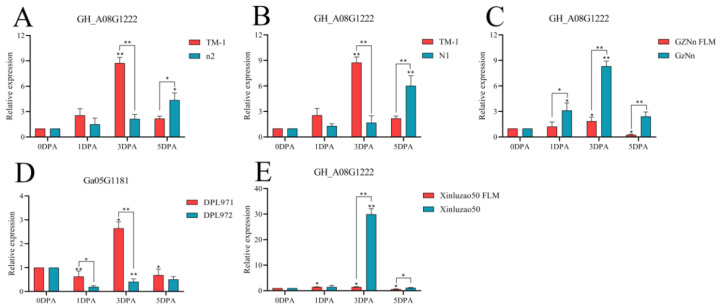
Relative expression levels of the *GH_A08G1222* gene in upland cotton. (**A**–**C**,**E**) Upland cotton; (**D**) homologous gene of Asian cotton. Asterisks at top of bars indicate statistically significant differences between periods (* *p* < 0.05, ** *p* < 0.01).

**Figure 18 genes-14-00208-f018:**
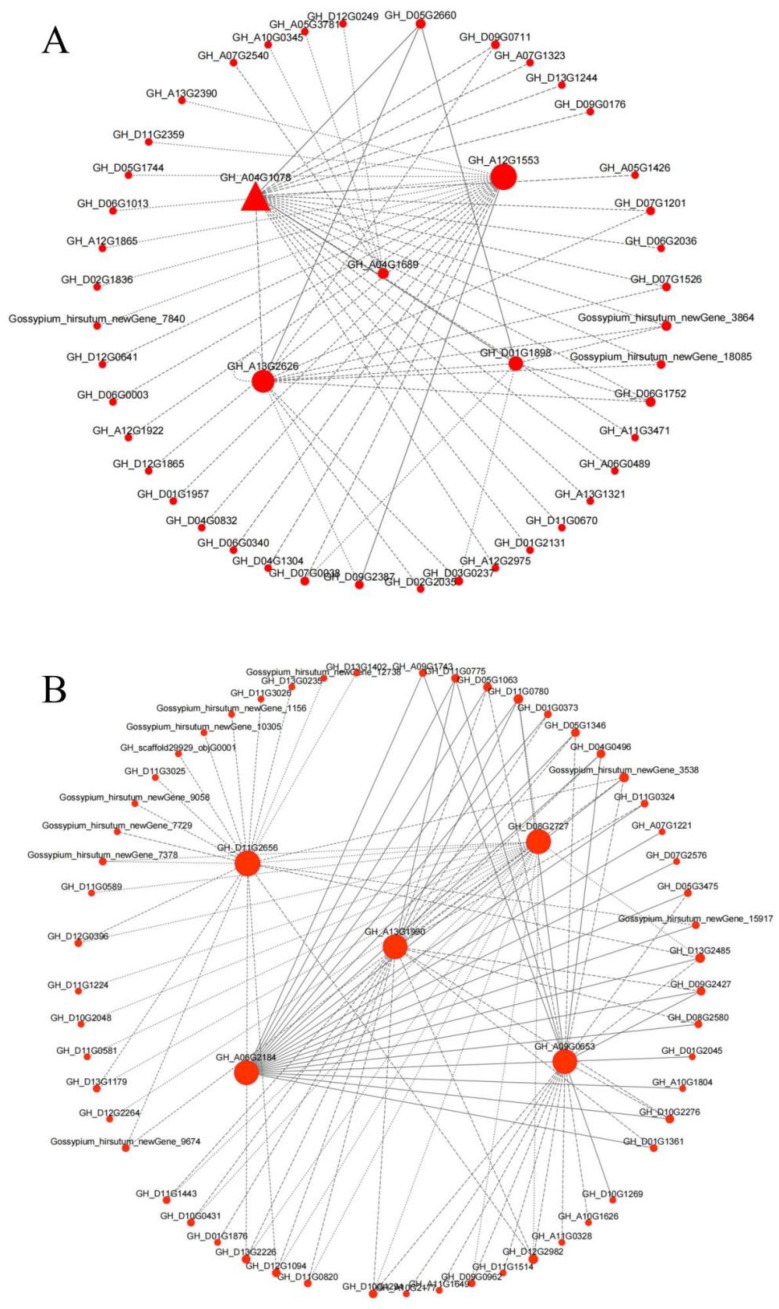
Gene co-expression network and hub genes in (**A**) MEred and (**B**) MEyellow modules.

## Data Availability

Not applicable.

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
