# Peer review of "Weighted Gene Co-Expression Network Analysis Reveals Hub Genes for Fuzz Development in *Gossypium hirsutum"

_genes, 2023, doi:10.3390/genes14010208_

Round 1
Reviewer 1 Report
Manuscript "Key genes and differential expression analysis of fiber development in Upland cotton fuzzless mutants" is very interesting.
General comments:
Authors explored the dynamic process and key genes of upland cotton fuzz initiation more effectively, and to provide a solid foundation for the molecular mechanism of upland cotton fuzz initiation. This is very important.
Unfortunately, description of statistical analysis is very poor.
Detailed comments:
Figures 2, 3: Authors used principal component analysis. This is incorrect for data with replications. Authors should use canonical variate analysis and estimate the Mahalanobis distances.
Quality of Figure 7 is very poor.
Figure 7 needs LSD or HSD values for comparison of mean values.
Figure 16 needs LSD or HSD values for comparison of mean values.
Figure 17 needs LSD or HSD values for comparison of mean values.
Figure 18 needs LSD or HSD values for comparison of mean values.
My suggestion:
Figures 2, 3. "PCA analysis", "A" - analysis. Repetition 'analysis'.
Figures 10, 11: lack information about values in parentheses.
Figure 5: "GO analysis" - more description.
Paper needs major revision.
Author Response
Response to Reviewer 1 Comments
Dear Professor: Thank you very much for your comments on the manuscript. We have taken your suggestions in consideration point by point and made the updating of this revision.
Point 1: Figures 2, 3: Authors used principal component analysis. This is incorrect for data with replications. Authors should use canonical variate analysis and estimate the Mahalanobis distances.
Response 1: Thank you so much for your suggestion. We modified in the manuscript.
Point 2: Quality of Figure 7 is very poor.
Figure 7 needs LSD or HSD values for comparison of mean values.
Figure 16 needs LSD or HSD values for comparison of mean values.
Figure 17 needs LSD or HSD values for comparison of mean values.
Figure 18 needs LSD or HSD values for comparison of mean values.
Response 2: Thank you so much for your suggestion. We modified in the manuscript and Attachment information.
Point 3: Figures 10, 11: lack information about values in parentheses.
Figure 5: "GO analysis" - more description.
Response 3:Thank you so much for your suggestion. We modified in the manuscript.
Thanks a lot for your patience and consideration.
Sincerely yours,
yanying Qu, PhD
Professor
Postal address: College of Agriculture, Xinjiang Agricultural University, 311 Nongda East Road, Urumqi, 830052, China.
Telephone number: 18599052306
Fax number: 0991-8762263
E-mail address:18599052306@163.com
Reviewer 2 Report
The manuscript lacks the novelty and written in very bad style and contains too many errors. The experimental design has serious flows.
The English language is very bad, and the manuscript is full of the errors, odd words, typos, spelling and grammatical errors.
The title is not suitable. I don’t see any importance for the words “key genes” in the title as the manuscript doesn’t involve any analysis or particular analysis for key pathways or key gene families.
The method section isn’t fully described. RNA extraction method and protocol lacks a lot of details about the used kit number or defining it correctly, RNA quality and integrity check, how is performed? A lot of details about cDNA synthesis, volume of RNA used ,…. etc.
The title of section 2.7.” Fluorescence quantitative QRT-PCR validation”, the word “fluorescence” is strange, and too many details are missed in this section as the volume of the PCR reaction and its components, Syber green kit or what?, PCR efficiency %, standard curve dilution series, …. Etc. Also, primers how they are designed by NCBI? How did you confirm its specificity to the selected genes? What are the reaction conditions and number of cycles?
The selected number of genes “eight” is too small for verification and analysis.
All the presented results are not clear, and not fully described with very short uninformative legend.
The resolution of Figures 5, 6, 7, 12 and 13 particularly is not good and writing on the figures is not obvious.
So, I must to reject it.
Author Response
Response to Reviewer 2 Comments
Dear Professor: Thank you very much for your comments on the manuscript. We have taken your suggestions in consideration point by point and made the updating of this revision.
Point 1: The title is not suitable. I don’t see any importance for the words “key genes” in the title as the manuscript doesn’t involve any analysis or particular analysis for key pathways or key gene families.
Response 1: Thank you so much for your suggestion. I have revised my title.
Point 2: The method section isn’t fully described. RNA extraction method and protocol lacks a lot of details about the used kit number or defining it correctly, RNA quality and integrity check, how is performed? A lot of details about cDNA synthesis, volume of RNA used ,…. etc.
Response 2: Thank you so much for your suggestion. We added a clarification to this,We modified in the manuscript.
Point 3: The title of section 2.7.” Fluorescence quantitative QRT-PCR validation”, the word “fluorescence” is strange, and too many details are missed in this section as the volume of the PCR reaction and its components, Syber green kit or what?, PCR efficiency %, standard curve dilution series, …. Etc. Also, primers how they are designed by NCBI? How did you confirm its specificity to the selected genes? What are the reaction conditions and number of cycles?
Response 3:Thank you so much for your suggestion. We had added a detailed explanation of this,We modified in the manuscript.
Point 4:The selected number of genes “eight” is too small for verification and analysis.
All the presented results are not clear, and not fully described with very short uninformative legend.
Response 4:Thank you so much for your suggestion. We re-analyzed these 8 genes and described them in detail.
Point 5:The resolution of Figures 5, 6, 7, 12 and 13 particularly is not good and writing on the figures is not obvious.
Response 5:Thank you so much for your suggestion. We renovated and modified the figures 5, 6, 7, 12 and 13 and uploaded them
Thanks a lot for your patience and consideration.
Sincerely yours,
yanying Qu, PhD
Professor
Postal address: College of Agriculture, Xinjiang Agricultural University, 311 Nongda East Road, Urumqi, 830052, China.
Telephone number: 18599052306
Fax number: 0991-8762263
E-mail address:18599052306@163.com
Reviewer 3 Report
The authors used the upland cotton fuzzless mutant to investigate the cotton fiber initiation 18 and development based on transcriptome sequencing, in particular, to explore the gene expression pattern and the differences between genes in upland cotton during fuzz period. The authors identified the genes likely involved in the formation of the fuzz of upland cotton. This study provided reference for molecular breeding and genetic improvement of cotton fiber.
I believe that the authors have provided sufficient background, explained well the methodology, and concluded appropriately based on available data. However, I strongly believe that the overall presentation of the manuscript needs to be significantly improved. I have listed the following issues (but I know there are more throughout the entire manuscript) for the authors to consider if a revision is requested by the editor.
Abstract: the abbreviations, e.g., GO, DEGs, etc., should be spelled with full names for their first use. It is so strange, there is always missing a space after the period “.” at the end of each sentence. I have also seen many other types of editorial errors, e.g., missing space, incorrect use of punctuations, etc. Please check and correct these problems for the entire manuscript.
Line 137: I believe that it is important to indicate the developmental stages of the ovules used to extract total RNA.
Figure 4: “A” and “B” on the figure are not matching “a” and “b” in the figure caption; this is the same problem in many other figures, please correct.
Figure 7: “A-H” need to be explained in the figure caption; the same problems in Figures 15-18.
Author Response
Response to Reviewer 3 Comments
Dear Professor: Thank you very much for your comments on the manuscript. We have taken your suggestions in consideration point by point and made the updating of this revision.
Point 1: Abstract: the abbreviations, e.g., GO, DEGs, etc., should be spelled with full names for their first use. It is so strange, there is always missing a space after the period “.” at the end of each sentence. I have also seen many other types of editorial errors, e.g., missing space, incorrect use of punctuations, etc. Please check and correct these problems for the entire manuscript.
Response 1: Thank you so much for your suggestion. We modified in the manuscript.
Point 2:Line 137: I believe that it is important to indicate the developmental stages of the ovules used to extract total RNA.
Response 2: We highly appreciate your suggestion. In this study, This data is public data downloaded from the Internet, not sequenced by myself. So I didn't include the extracted total RNA
Point 3: Figure 4: “A” and “B” on the figure are not matching “a” and “b” in the figure caption; this is the same problem in many other figures, please correct.
Response 3: Thank you so much for your suggestion. We modified in the manuscript.
Point 4: Figure 7: “A-H” need to be explained in the figure caption; the same problems in Figures 15-18.
Response 4: Thank you so much for your suggestion. We modified in the manuscript.
Thanks a lot for your patience and consideration.
Sincerely yours,
yanying Qu, PhD
Professor
Postal address: College of Agriculture, Xinjiang Agricultural University, 311 Nongda East Road, Urumqi, 830052, China.
Telephone number: 18599052306
Fax number: 0991-8762263
E-mail address:18599052306@163.com
Round 2
Reviewer 1 Report
First round:
Point 1: Figures 2, 3: Authors used principal component analysis. This is incorrect for data with replications. Authors should use canonical variate analysis and estimate the Mahalanobis distances.
Authors' Response 1: Thank you so much for your suggestion. We modified in the manuscript.
Figures 2 and 3: Authors did not improve manuscript. Still is PCA. This is incorrect.
Figure 6 still needs LSD or HSD values for comparison of mean values.
Figure 14 still needs LSD or HSD values for comparison of mean values.
Figure 15 still needs LSD or HSD values for comparison of mean values.
Figure 16 still needs LSD or HSD values for comparison of mean values.
Figure 17 still needs LSD or HSD values for comparison of mean values.
First round:
Figure 5: "GO analysis" - more description.
Response 3:Thank you so much for your suggestion. We modified in the manuscript.
Still lack of description of GO analysis.
Paper needs major revision.
Author Response
Response to Reviewer 1 Comments
Dear Professor: Thank you very much for your comments on the manuscript. We have taken your suggestions in consideration point by point and made the updating of this revision.
Point 1: Point 1: Figures 2, 3: Authors used principal component analysis. This is incorrect for data with replications. Authors should use canonical variate analysis and estimate the Mahalanobis distances.
Response 1: Thank you so much for your suggestion. I see other people doing PCA in this way, so I think my method is correct.
Point 2: Figure 6 still needs LSD or HSD values for comparison of mean values.
Figure 14 still needs LSD or HSD values for comparison of mean values.
Figure 15 still needs LSD or HSD values for comparison of mean values.
Figure 16 still needs LSD or HSD values for comparison of mean values.
Figure 17 still needs LSD or HSD values for comparison of mean values.
Response 2: Thank you so much for your suggestion. For Figure 6, I see others doing the same analysis using my method to verify the reliability of the transcriptome data. For Figure 14-17, I have analyzed it with LSD analysis method. Please refer to the revised draft and attachment for details.
Point 3: Still lack of description of GO analysis..
Response 3:Thank you so much for your suggestion. We modified in the manuscript and attachment for details.
Thanks a lot for your patience and consideration.
Sincerely yours,
yanying Qu, PhD
Professor
Postal address: College of Agriculture, Xinjiang Agricultural University, 311 Nongda East Road, Urumqi, 830052, China.
Telephone number: 18599052306
Fax number: 0991-8762263
E-mail address:18599052306@163.com
Reviewer 2 Report
The authors of the manuscript with ID no. "genes-1881970" didn't improve the manuscript and didn't provide point-by-point response to the reviewer's comment. I revised the manuscript and checked points that I did comments on it and I found the authors only changed the title of the manuscript and did minor edits to the English language. The English language is still not good enough, the presentation of the results is not clear and figures' resolution is too bad, the figure's captions are too summarized and not informative. furthermore, the methods are not fully described and authors didn't respond to the comments related to this issue. In most of case, the results were very primary, and just were list, or not further investigated.
So, I see that the manuscript is not suitable for publication in Genes Journal.
Author Response
Response to Reviewer 2 Comments
Dear Professor: Thank you very much for your comments on the manuscript. We have taken your suggestions in consideration point by point and made the updating of this revision.
Point 1: The title is not suitable. I don’t see any importance for the words “key genes” in the title as the manuscript doesn’t involve any analysis or particular analysis for key pathways or key gene families.
Response 1: Thank you so much for your suggestion. I have revised my title.I have changed the title of my article from "key genes" to "hub genes".
Point 2: The method section isn’t fully described. RNA extraction method and protocol lacks a lot of details about the used kit number or defining it correctly, RNA quality and integrity check, how is performed? A lot of details about cDNA synthesis, volume of RNA used ,…. etc.
Response 2: Thank you so much for your suggestion. We added a clarification to this,We modified in the manuscript.121/5000
We have added details such as: RNA quality and integrity check, how is performed? A lot of details about cDNA synthesis, volume of RNA used
Point 3: The title of section 2.7.” Fluorescence quantitative QRT-PCR validation”, the word “fluorescence” is strange, and too many details are missed in this section as the volume of the PCR reaction and its components, Syber green kit or what?, PCR efficiency %, standard curve dilution series, …. Etc. Also, primers how they are designed by NCBI? How did you confirm its specificity to the selected genes? What are the reaction conditions and number of cycles?
Response 3:Thank you so much for your suggestion. We had added a detailed explanation of this,We modified in the manuscript.We have added details
Point 4:The selected number of genes “eight” is too small for verification and analysis.
All the presented results are not clear, and not fully described with very short uninformative legend.
Response 4:Thank you so much for your suggestion. We did this analysis only to verify the accuracy of transcriptome data, so we selected eight genes. We've seen other similar papers that have selected eight genes. We re-analyzed these 8 genes and described them in detail.
Point 5:The resolution of Figures 5, 6, 7, 12 and 13 particularly is not good and writing on the figures is not obvious.
Response 5:Thank you so much for your suggestion. We renovated and modified the figures 5, 6, 7, 12 and 13 and uploaded them.Please see the article or the attachment
Thanks a lot for your patience and consideration.
Sincerely yours,
yanying Qu, PhD
Professor
Postal address: College of Agriculture, Xinjiang Agricultural University, 311 Nongda East Road, Urumqi, 830052, China.
Telephone number: 18599052306
Fax number: 0991-8762263
E-mail address:18599052306@163.com
Round 3
Reviewer 1 Report
Figures 2 and 3: Authors did not improve manuscript. Still is PCA. This is incorrect. Authors should use canonical variate analysis and estimate the Mahalanobis distances!
Figure 6 still needs LSD or HSD values for comparison of mean values!
Figure 14 still needs LSD or HSD values for comparison of mean values!
Figure 15 still needs LSD or HSD values for comparison of mean values!
Figure 16 still needs LSD or HSD values for comparison of mean values!
Figure 17 still needs LSD or HSD values for comparison of mean values!
Still lack of description of GO analysis!
Author Response
Response to Reviewer 1 Comments
Dear Professor: Thank you very much for your comments on the manuscript. We have taken your suggestions in consideration point by point and made the updating of this revision.
Point 1: Point 1: Figures 2, 3: Authors used principal component analysis. This is incorrect for data with replications. Authors should use canonical variate analysis and estimate the Mahalanobis distances.
Response 1: The traditional PCA analysis method used in Figures 2 and 3 of the author's paper was used to evaluate the consistency of transcriptome data, which was also used in other studies. I don't see a similar article using canonical variate analysis and estimating the Mahalanobis distances. Therefore, I did not modify it. Please see what is wrong with this analysis..
Point 2: Figure 6 still needs LSD or HSD values for comparison of mean values.
Figure 14 still needs LSD or HSD values for comparison of mean values.
Figure 15 still needs LSD or HSD values for comparison of mean values.
Figure 16 still needs LSD or HSD values for comparison of mean values.
Figure 17 still needs LSD or HSD values for comparison of mean values.
Response 2: Thank you so much for your suggestion. For Figure 6, I see others doing the same analysis using my method to verify the reliability of the transcriptome data. For Figure 14-17, I have analyzed it with LSD analysis method. Please refer to the revised draft and attachment for details.
Point 3: Still lack of description of GO analysis..
Response 3:Thank you so much for your suggestion. We modified in the manuscript and attachment for details.
Thanks a lot for your patience and consideration.
Sincerely yours,
yanying Qu, PhD
Professor
Postal address: College of Agriculture, Xinjiang Agricultural University, 311 Nongda East Road, Urumqi, 830052, China.
Telephone number: 18599052306
Fax number: 0991-8762263
E-mail address:18599052306@163.com